# Comparative Transcriptome Analysis Reveals the Innate Immune Response to *Mycoplasma gallisepticum* Infection in Chicken Embryos and Newly Hatched Chicks

**DOI:** 10.3390/ani13101667

**Published:** 2023-05-17

**Authors:** Mengyun Zou, Tengfei Wang, Yingjie Wang, Ronglong Luo, Yingfei Sun, Xiuli Peng

**Affiliations:** Key Laboratory of Agricultural Animal Genetics, Breeding and Reproduction, Ministry of Education, College of Animal Science and Technology and College of Veterinary Medicine, Huazhong Agricultural University, Wuhan 430070, China

**Keywords:** *Mycoplasma gallisepticum*, immune response, chicken embryo, chick, RNA-seq

## Abstract

**Simple Summary:**

*Mycoplasma gallisepticum* causes respiratory disease in chickens. This study used RNA-seq to investigate the immune response of chicken embryos and chicks to infection. Weight loss and immune damage were observed in infected chickens of both ages. Transcriptome analysis revealed differentially expressed genes related to innate immunity and inflammation, with toll-like receptors and cytokine pathways playing a dominant role in the immune response. The immune response was stronger in embryos than in chicks. These findings provide valuable insights for developing disease control strategies.

**Abstract:**

*Mycoplasma gallisepticum* (*MG*) is a major cause of chronic respiratory diseases in chickens, with both horizontal and vertical transmission modes and varying degrees of impact on different ages. The innate immune response is crucial in resisting *MG* infection. Therefore, this study aimed to investigate the innate immune response of chicken embryos and newly hatched chicks to *MG* infection using comparative RNA-seq analysis. We found that *MG* infection caused weight loss and immune damage in both chicken embryos and chicks. Transcriptome sequencing analysis revealed that infected chicken embryos had a stronger immune response than chicks, as evidenced by the higher number of differentially expressed genes associated with innate immunity and inflammation. Toll-like receptor and cytokine-mediated pathways were the primary immune response pathways in both embryos and chicks. Furthermore, TLR7 signaling may play an essential role in the innate immune response to *MG* infection. Overall, this study sheds light on the development of innate immunity to *MG* infection in chickens and can help in devising disease control strategies.

## 1. Introduction

*Mycoplasma gallisepticum* (*MG*) is the major cause of chronic respiratory disease (CRD) in chickens. Infection with MG leads to slow growth, poor feed conversion, and reduced hatchability in chickens, thus bringing huge economic losses to the poultry industry [1,2]. Notably, *MG* can not only spread horizontally through the environment but also vertically to the next generations, which is a primary consideration for international trade [3]. Vaccination and antibiotics are currently effective methods to control *MG* infection. However, the available vaccines and antibiotics all have limitations due to high strain antigen variability, drug residues, and the emergence of resistant strains [1,4]. A thorough understanding of the molecular mechanism of the host immune system against *MG* infection is necessary to explore new prevention and treatment strategies for *MG*.

Transcriptome sequencing has great advantages in providing the host immune response to microbial infections. As shown in Table 1, there is growing evidence from transcriptome sequencing that *MG* infection is characterized by vigorous inflammatory responses and dysregulated immune responses in the respiratory tract of chickens. Recent findings not only reaffirm these results but also suggest that the immune protection provided by *MG* vaccine strains was achieved in part by balancing cytokines and chemokines production and TLRs expressions in chickens [5]. Due to high reproducibility and well-known biology and physiology, the chicken embryo is an important model for studying the development of innate immunity and vaccine efficacy [6,7]. Increasingly, studies have shown that the in ovo administration of immune modulators can enhance the innate immunity of chickens in adulthood [8,9]. However, the regulatory pathways and molecular mechanisms underlying the immune response induced by *MG* in chicken embryos have been rarely explored.

Hence, this study aimed to explore the genes and pathways underlying the *MG*-induced pathological and immune events in newly hatched chicks and chicken embryos’ lungs. Alveolar epithelial cells (AECII) are important functional and structural cells in the alveoli. They can differentiate into AECI cells and secrete cytokines and chemokines, which are essential for maintaining the normal function of lung tissue and regulating the lung immune balance [10]. Therefore, we also verified the transcriptome sequencing results in primary AECII cells. Results reveal the transcriptional responses of chicks and chicken embryos’ lungs in defense against *MG* and detail the involved immune-related genes and pathways in the transcriptome. Understanding the differences in the immune response to *MG* infection between embryonic chickens and chicks will help better develop disease control strategies.

**Table 1 animals-13-01667-t001:** Summary of transcriptome expression in response to *MG* infection in chicken tracheal and lung tissues.

Animal	Sample	Groups	Pathways	Reference
5-week SPF female chicken	Tracheal epithelial cells	CG vs. *M*G R_low_, R_high_, R_low_ LAMP, and R_high_ LAMP	Up: TLRs, TNF-α/NF-kB, adipogenesis, senescence and autophagy, MAPK, apoptosis, EGFR1, Type II interferon	[11]
5-week SPF chicken	1, 3, 5, and 7 days after challenge Tracheal	CG vs. *MG* Rlow	Up: apoptosis, TLRs, MAPK, JAK-SAT, NKC mediated cytotoxicity, CCRI, intestinal immune network for IgA production	[12]
5-week SPF chicken	1 and 2 days after infection Tracheal lumen cell	CG vs. *MG* Rlow, GT5, and Mg7	Up: CCRI, MAPK, NOD receptor, JAK-SAT, TLRs	[13]
5-week SPF chicken	6 days after infection Tracheal rings	CG/H3N8 vs. *MG*R_low_/H3N8	Up: phagosome, cell adhesion molecules, intestinal immune network for IgA production, JAK/STAT, TLRs Down: cardiac muscle contraction, purine metabolism, oxidative phosphorylation	[14]
10d chick	3 days after infection Lung tissue	CG vs. *MG* Rlow/*E. coli* O78	Up: CCRI, phagosome, IL-17, NF-κB, cell adhesion molecules	[15]
70-week or 4-week SPF chicken	Two weeks after challenge Tracheal tissues	CG vs. *MG* Asp3A	Up: DNA replication, cell cycle, CCRI, TLRs, phagosome, Down: formation and motor movement of the cilia, adherens junctions, tight junctions	[5]
3-week SPF chicken	15 days after infection Tracheal	CG vs. MG-HY	Up: cell adhesion molecules, phagosome, MAPK, calcium, and PPAR Down: Tight, ECM-receptor interaction, focal, AMPK, p53, Rap1, regulation of actin	[16]

## 2. Materials and Methods

### 2.1. Ethics in Animal Experimentation

Our experimental protocols for animals were authorized by the Ethical Committee on Animal Research at Huazhong Agricultural University (HZAUCH-2020-0003) and met the International Guiding Principles for Biomedical Research Involving Animals as issued by the Council for the International Organizations of Medical Sciences.

### 2.2. Mycoplasma gallisepticum Strain and Culture

The HS strain of *MG* used in the study is a pathogenic strain that was isolated from a local chicken farm in Hubei province in 1988. The culture medium was FM-4 medium supplemented with 12% inactivated porcine serum, 1% phenol red, and penicillin sodium (10,000 IU/mL), as previously described [17]. *MG* was inoculated in FM-4 medium and cultured at 37 °C with 5% CO_2_ until the medium color changed from red to orange. Hemagglutination and color-changing unit assays were used to determine the virulence of MG and the concentration of *MG* per milliliter of medium.

### 2.3. Cell Culture and Treatment

Primary chicken alveolar type II cells (AEC-II) were obtained from 13.5–15 embryonic day (ED) chickens. The method of isolation and preparation for AEC-II has been described previously [18]. In this study, AEC-II cells were seeded at a density of 1 × 10^6^ into six-well plates and cultivated at 37 °C with 5% CO_2_. When the cell confluence reached 60–70%, the *MG* group cells were treated with 200 uL of *MG* (1 × 10^9^ CCU/mL), and the control group cells were treated with the same volume of PBS. After 6 h of infection, the total RNA of treated cells was collected for further study.

### 2.4. Experimental Infection of Chicks and Chicken Embryos

The methods and procedures for establishing the *MG*-infected specific-pathogen-free (SPF) chicken embryo model have been described previously [19]. Briefly, 300 1-day-old SPF fertilized eggs were purchased from Merial Vital Laboratory Animal Technology (Beijing, China) and incubated to 9 days of age in biochemical incubators with a relative humidity of 55–60% and temperature of 37.8 ± 0.1 °C. On the ninth day of incubation, 180 embryos were treated with 800 μL of *MG*-HS at 1 × 10^9^ CCU/mL through the allantoic cavity injection. The others were injected with the same volume of saline as controls. Three to 11 days post-infection, 12 embryonic eggs were randomly selected from each group every day for body weight measurement and tissue isolation. To determine the immune organ index, fresh spleen tissue, bursa of Fabricius tissue, and thymus tissue were weighed immediately.

The methods and procedures for establishing the *MG*-infected chick model have been described previously [2]. Briefly, 50 one-day-old SPF fertilized eggs were purchased from Merial Vital Laboratory Animal Technology (Beijing, China) and incubated until hatching in a sterilized biochemical incubator with a relative humidity of 55–60% and temperature of 37.8 ± 0.1 °C. Thirty-five newborn chicks were reared for 10 days in a separate sterile brood room. At 10 days old, 35 chicks were randomly divided into 2 groups with 3 replicates and 5 chicks each replicate. The chicks in the infected group were treated with 1 mL *MG*-HS at 1 × 10^9^ CCU/mL through the left air sacs, nose, and eye. The others were injected with the same volume of saline as controls. At 5 days post-infection, the chicks in the infected group all showed clinical symptoms of rales, mouth breathing, coughing, and sneezing and were humanely euthanized for body weight detection and tissue collection. Similarly, to determine the immune organ index, fresh spleen tissue, bursa of Fabricius tissue, and thymus tissue were weighed immediately. All isolated tissue was immediately stored in liquid nitrogen for further experiments.

### 2.5. Total RNA Extraction and Real-Time Quantitative PCR (RT-qPCR)

The total RNA from the lungs, spleen, bursal, and thymus was collected by TRIzol reagent (Invitrogen, Waltham, MS, USA) according to the manufacturer’s instructions. The RNA integrity and quality were monitored on 1% agarose gels and assessed using the RNA Nanodrop2000 (Thermo Scientific, CA, USA) machine. Then, 1 μg total RNA was reverse transcribed into cDNA using the first strand cDNA synthesis kit (TaKaRa, Tokyo, Japan). Subsequently, RT-qPCR was used to determine the relative mRNA expression of specific genes using TransStart Top Green qPCR SuperMix (YEASEN, Shanghai, China). The relative expression levels of genes were normalized to GAPDH. All DNA primer sequences are shown in Table 2.

### 2.6. Illumina Sequencing

Magnetic beads containing Oligo(dT) were first used to enrich mRNA from total RNA. Then mRNA was used as a template to synthesize single-stranded cDNA and then double-stranded cDNA. The cDNA was then purified and subjected to end repair, A-tailed, and ligated with sequencing adapters, and then AMPure XP beads were used for fragment size selection. Finally, PCR amplification was performed, and the PCR products were purified to obtain the final cDNA library. Qubit 2.0 was used for preliminary quantification. The Illumina high-throughput sequencing platform NovaSeq 6000 (Illumina, CA, USA) was used for sequencing to generate 150 bp paired-end reads.

### 2.7. Transcriptome Data Analysis

The low-quality and adaptor sequences were removed from the raw data using the Sequence CLEAN program to obtain clean data (Fastq files). After quality control with fastaQC (version 0.11.7), the sequences were mapped to the *Gallus gallus* genome assembly with Hisat2 (Gallus genome Version 5.0.1 NCBI) [20]. DESeq2 was used to determine DEGs according to default criteria consisting of a |log2 Fold Change (FC)| > 1.5 and an adjust-*p* value < 0.05. Functional analysis of DEGs was performed using the Gene Ontology (GO), Kyoto Encyclopedia of Genes and Genomes (KEGG) pathway, and Reactome pathway databases. All data represent the average change in gene expression from three independent replicates. In this study, GO and pathway terms enriched at an FDR < 0.05 were reported and considered significant.

### 2.8. Histopathological Evaluation of Trachea, Lungs, and Immune Organs in Birds

The trachea, lung, spleen, bursa of Fabricius, and thymus of 3 birds of each group were histopathologically evaluated using hematoxylin-eosin (H&E) staining, which has been described previously [21].

### 2.9. Statistical Analysis

IBM SPSS Statistics 19 software (Armonk, NY, USA) was used to analyze the data. The differences between groups were analyzed by one-way ANOVA using Duncan’s multiple-range test. The results are presented as the mean ± SD. Each experiment group has at least three samples. A value of *p* < 0.05 was statistically significant.

## 3. Results

### 3.1. MG Infection Leads to Reduced Body Weight and Dysregulated Immune Organs Index in Chicken Embryos and Newly Hatched Chicks

To evaluate the effects of *MG* infection on the chickens’ growth and immunity, the body weight and immune organ weights of two birds were weighed. As shown in Figure 1a, the body weight of infected chicken embryos at 3, 5, 6, 9, 10, and 11 day(s)post-infection (equivalent to 12, 14, 15, 18, 19, and 20 day(s) of hatching) was decreased significantly relative to the control group. As shown in Figure 1b, *MG* infection resulted in an obvious reduction in the weight gain of newly hatched chicks compared with the non-infection group. What’s more, the spleen index of chicken embryos at 7, 8-, 9-, 10-, and 11-day post-infection (Figure 1c), the thymus index of chicken embryos at 6 d post-infection (Figure 1d), and the bursal index at 9 days post-infection were remarkably elevated in the infection group, while the bursal index at 5- and 11-day(s) post-infection were decreased in the infection group relative to the control group (Figure 1e). In addition, the spleen and bursal indexes of infected chicks were significantly elevated relative to the control group, while the thymus index of infected chicks was reduced (Figure 1f). Taken together, *MG* infection reduces body weight in chickens and embryos and leads to abnormal immune organ indexes in chicks and chicken embryos.

### 3.2. MG-Induced Histopathological Damage in Chicks and Chicken Embryos

Next, we further investigated the histopathological effects of *MG* infection on the respiratory tract and immune organs of two birds. As shown in Figure 2a,b, no pathological changes were found in the lungs and trachea of chicken embryos in the non-infection group. In contrast, after *MG* infection, the trachea in chicken embryos showed inflammatory infiltration and increased mucosal tracheal thickness (Figure 2a), and the lungs showed narrowed alveolar space and cell shedding (Figure 2b). As shown in Figure 2c,d, the lung and trachea in the non-infection group showed normal and intact structure. However, *MG* infection resulted in significant lymphocytes infiltration, increased trachea mucosa thickness, and shedding of ciliated epithelial cells in chick trachea (Figure 2c) and narrowed alveolar space and severe inflammatory lesions in chick lung tissue (Figure 2d).

Furthermore, obvious pathological changes were also found in the immune organs of infected chicks and chicken embryos. As indicated in Figure 2e,f, non-infected embryonic spleen and bursa showed normal morphology, while infected embryonic spleen and bursa showed lymphocyte reduction and interstitial edema. Similar results were also observed in chicks. As shown in Figure 2g–i, the non-infected chick spleen, thymus, and bursa showed normal and complete structure without any histopathological changes. In contrast, infected chick spleens showed enlarged splenic nodules, inflammatory factor infiltration, lymphocyte reduction, and irregular arrangement of cells (Figure 2g); infected chick thymus and bursa of Fabricius all exhibited inflammatory cell infiltration and lymphocytes reduction (Figure 2h,i). In brief, *MG* infection induced immune damage in the respiratory tract and immune organs of both chicks and chicken embryos.

### 3.3. Transcriptome Sequencing Analysis of Chicks and Chicken Embryos Lung Tissues

Illumine deep sequencing was applied to assess the global transcripts from the lungs of chicks and chicken embryos with *MG* infection. At an adjust-*p* value < 0.05, 389 DEGs were found in the lungs of infected chicken embryos compared to the control group, with much more upregulated genes (374/389, 96.14%) than downregulated genes (15/374, 3.86%) (Figure 3a). The top 30 upregulated and all 15 down-regulated genes in the lungs of infected chicken embryos were listed in Appendix A, respectively. Meanwhile, 417 DEGs were found in the lungs of infected chicks compared to the control group, among which the up-regulated genes (315/417 75.53%) were more than the down-regulated genes (102/417, 24.46%) (Figure 3b). The top 30 upregulated and downregulated genes in the lungs of infected chicks were listed in Appendix A, respectively. Through the Venn diagram, we found that there were 287 chicken embryo-specific DEGs, 229 chick-specific DEGs, and 86 shared DEGs (Figure 3c). Of the overlapped DEGs, all 86 DEGs were up-regulated in *MG*-infected chick and chicken embryo lungs. The top 30 overlapped up-regulated genes are listed in Table 3.

### 3.4. GO Analysis of the DEGs in Chicken Embryos and Chicks after MG Infection

GO analysis of DEGs enabled the genes to be categorized based on enriched cellular components (CCs), molecular functions (MFs), and biological processes (BPs). At an FDR of <0.05, 374 upregulated genes in infected chicken embryo lungs were assigned to 3, 14, and 53 GOs for CCs, MFs, and BPs, respectively. The enriched CCs and MFs for upregulated genes are shown in Appendix A, and the top 20 significant enriched GOs for upregulated genes under BPs are shown in Figure 4a. The top three CCs enriched with upregulated genes were mainly associated with extracellular space, the external side of the plasma membrane, and the integral component of the membrane (Appendix A); the top three MFs enriched with upregulated genes were mainly associated with chemokine activity, CCR6 chemokine receptor binding, and CCR chemokine receptor binding (Appendix A), and the top 3 BPs enriched with upregulated genes were mainly associated with inflammatory response, immune response, neutrophil chemotaxis (Figure 4a). At an FDR < 0.05, 315 upregulated genes in infected chick lungs were assigned to 7, 10, and 21 GOs for CCs, MFs, and BPs, respectively, while downregulated genes were assigned to 3, 0, and 0 GOs for CCs, MFs, and BPs, respectively. The enriched CCs and MFs for up-and down-regulated genes are shown in Appendix A, and the top 21 enriched BPs for up-regulated genes are shown in Figure 4b. The top three CCs enriched with upregulated genes were mainly associated with chromosome, MCM complex, and nucleoplasm (Appendix A), the top 3 MFs enriched with upregulated genes were associated with ATP-dependent microtubule motor activity, transmembrane signaling receptor activity, and ATP binding (Appendix A), and the top three BPs enriched with upregulated genes were involved in inflammatory response, cell division, and neutrophil chemotaxis (Figure 4b). The CCs enriched with downregulated genes were associated with myelin sheath, haptoglobin–hemoglobin complex, and hemoglobin complex. Taken together, the co-activation of inflammatory response, immune response, and neutrophil chemotaxis in *MG*-infected chicken embryos and chick lung tissue suggests that these biological processes play a dominant role in the response of chickens to *MG* infection.

In order to further explore the similarities and differences in immune responses between chicken embryos and chick lungs after *MG* infection, we calculated the number of DEGs involved in immune processes. We found that 75.84% of DEGs (295/389) in infected chick embryo lung tissue were associated with immune responses, which was significantly higher than that of 13.19% (55/417 DEGs) in chicks, indicating that the immune response of *MG*-infected chicken embryos was stronger than that of chicks.

### 3.5. Pathway Analysis of the DEGs in Chicks and Chicken Embryos after MG Infection

At an FDR of <0.05, there are 10 KEGG pathways enriched in the upregulated genes in infected chicken embryo lungs relative to the control group, but no KEGG pathway was enriched in the downregulated genes. The most significantly enriched pathway was the cytokine–cytokine receptor interaction (CCRI) pathway, followed by the toll-like receptor (TLR) pathway, phagosome, and NOD-like receptor, influenza A, herpes simplex virus 1 infection, intestinal immune network for IgA production, cytosolic DNA-sensing pathway, cell adhesion molecules, and salmonella infection signaling pathways (Figure 5a). At an FDR < 0.05, there were 5 KEGG pathways enriched in the upregulated genes in infected newly hatched chicks’ lungs, while no KEGG pathways were enriched with downregulated genes. The most significantly enriched pathway was also the CCRI pathway, followed by cell cycle, DNA replication, intestinal immune network for IgA production, and TLR signaling pathway (Figure 5b). The summary diagram based on GO and KEGG enrichment items is shown in Figure 6 and Figure 7.

### 3.6. Validation of CCRs in Both Chicks and Chicken Embryos after MG Infection

As the CCRI pathway was the most enriched pathway in both infected chicken embryos and newly hatched chicks, we further analyzed DEGs enriched in this pathway. According to the KEGG pathway analysis, there are 34 and 22 DEGs enriched in the CCRI pathways in infected chicken embryos and chick lung tissues, respectively (Figure 8a). To validate the sequencing result, eight common DEGs, including CCR2, CSF3R, TNFRSF6B, CCL4, IL20RA, IL8, CXCL13, and CXCL13L2 and one added chick-unique gene MMP7, were selected for further analysis. Transcriptome sequencing results showed that these eight CCRI genes were upregulated by 4.64 times in chicken embryo lung tissue (Figure 8b) and 3.18 times on average in that of chicks after *MG* infection (Figure 8d). The RT-qPCR results showed that, except for the CXCL13 and MMP7, all shared seven genes were upregulated by about 12.4-fold on average in chicken embryo lungs after infection (Figure 8c), while in *MG*-infected chick lung tissue, all eight genes were upregulated by about 5.4-fold on average, except for IL20RA (Figure 8e). The results of RT-qPCR were basically consistent with the sequencing results except for the different multiples of gene expression changes, which confirmed the reliability of the sequencing results. In addition, the results of sequencing and RT-qPCR showed that the number and multiple of upregulated CCRI genes in chicken embryos after *MG* infection were higher than those in chicks.

### 3.7. Validation of TLRs in Both Chick and Chicken Embryo after MG Infection

Since DEGs were significantly enriched in the TLRs pathway in the lung tissues of chicks and chick embryos infected with *MG*, we further analyzed all avian TLRs in the lungs and immune organs. The RNA-seq results showed that all avian TLRs, except for TLR5 and TLR21, were upregulated by 4.79-fold on average in infected chicken embryos’ lungs compared to the control group (Figure 9a), while only TLR1A and TLR7 were upregulated by about 2.05-fold on average in infected chick lungs compared to the control group (Figure 9c). RT-qPCR results showed that the expressions of all avian TLRs, except for TLR5 and TLR21, were evidently increased by about 6-fold on average in infected embryonic lungs (Figure 9b). However, in infected chick lungs, TLR1A, 2, 6, 7, and 21 expressions were significantly increased by 1.75-fold on average, TLR5 expression was decreased by about 0.5-fold, and for TLR3, four expressions showed no significant change (Figure 9d). These results were generally consistent with the sequencing results, indicating the reliability of the sequencing data.

To further understand the roles of TLRs in the host immune system, we further examined the expressions of these TLRs in immune organs after *MG* infection. The results in chicken embryos showed that TLRs were expressed differently in different tissues, and the overall expression of TLRs in the spleen was less changed than that in the lungs, thymus, and bursa of Fabricius. In the infected embryonic spleens, TLR1A, 2, 6, and 15 were significantly upregulated, while TLR3, 4, 5, 7, and 21 had no significant changes (Figure 9e). However, in the infected embryonic thymus, TLR2, 4, 5, 6, and 15 were highly expressed, while TLR1A, 3, 7, and 21 had no significant changes (Figure 9f). In infected embryonic bursa of Fabricius, TLR2, 4, 5, 6, and 15 were highly elevated relative to the control group, while TLR1A, 3, 7, and 21 had no significant changes (Figure 9g).

The results in chicks also showed that the overall expression of TLRs in the spleen was less changed than that in the lungs, thymus, and bursa of Fabricius after *MG* infection. In infected chick spleen, except for the upregulation of TLR6 and TLR21 and downregulation of TLR5, the expression of most TLRs did not change significantly (Figure 9h). However, in infected chick thymus, all the TLRs except TLR15 and TLR21 were upregulated relative to the control group (Figure 9i). In infected chick bursa, only TLR4, 6, 7, and 21 were highly expressed, but TLR1A, 15, 5, 2, and 3 had no significant changes compared to the control group (Figure 9j).

### 3.8. Analysis of Protein Interaction Network

To find out the core immune genes in regulating the immune response to *MG* infection, we first analyzed the 86 DEGs shared by infected chicken embryos and newly hatched chicks through GO and pathway analysis. At an FDR of <0.05, these DEGs were assigned to 0, 3, and 13 GOs for CCs, MFs, and BPs, respectively. Consistently, the three MFs were chemokine activity, CXCR chemokine receptor binding, and transmembrane signaling receptor activity, and the top five BPs enriched with upregulated genes were neutrophil chemotaxis, inflammatory response, chemokine-mediated signaling pathway, and antimicrobial humoral immune response (Figure 10a). KEGG pathway analysis showed that activation of the CCRI and TLRs signaling pathways was conserved in the immune response of chick embryos and chicks to *MG* infection (Figure 10b). The similarities and differences in immune responses of chicks and chicken embryos to *MG* infection are outlined in Table 4.

Protein interaction network analysis through STRING showed that 10 genes, including but not limited to TLR7, S100A9, IL8L1, and IL1B, were in the center of the interaction network, indicating that these genes may play dominant roles in response to *MG* infection in both chicken embryos and chicks (Figure 10c). By comprehensively analyzing the immune genes significantly associated with *MG* infection reported by previous studies and the core genes found in this study, we identified 11 immune genes through the Venn diagram, including CXCL13L2, CXCL13, IL8L2, IL8L1, IL4I1, AVD, IL1B, TLR7, S100A9, IL13RA2, and MMP7, which may be the core immune genes regulating *MG* infection. STRING analysis showed that IL1B, IL8L1, IL8L2, and CXCL13L2 were at the center of network nodes (Figure 10d). Consistently, RT-qPCR results showed that these genes were significantly upregulated in *MG*-infected AEC-II cells (Figure 10e).

To further explore whether these genes are related to *MG* loading, we treated AEC-II cells with different concentrations of *MG*. The results showed that, with the increase of *MG* infection concentration, the expressions of CXCL13L2, IL1B, TLR7, IL8L2, IL8L1, and IL4I1 were in dynamic changes; AVD and S100A9 were increased in an *MG* concentration-dependent manner; MMP7 was significantly up-regulated in low *MG* treatment, and down-regulated with the increase of *MG* treatment concentration (Figure 10f).

## 4. Discussion

Many researchers have shown that *MG* infection is characterized by airway inflammation and dysregulated immunity. The findings of this study not only affirmed this view but also revealed that infected chicken embryos have higher levels of immune response in quantity and quality than infected chicken embryos.

Innate immunity against respiratory pathogens involves physical and immunological mechanisms. *MG* infection induced immune damage and apoptosis in the respiratory tract and immune organs of chickens [2,21].

The immune system of chickens begins to develop during the embryonic stage and continues to mature after hatching. During the embryonic development of chickens, the first lymphoid organ to develop is the thymus, then the bursa of Fabricius, and finally, the spleen. At the embryonic stage, the yolk sac and the embryonic bursa of Fabricius are the primary sites of immune cell development. As the chicken grows, the bone marrow and thymus become the primary sites of immune cell production. During embryo development, the spleen has a high contribution to lymphopoiesis.; however, it only becomes a secondary lymphoid organ once the egg hatches [7]. The spleen is a peripheral immune organ in birds, while the bone marrow, bursa, and thymus are considered central immune organs [22]. Damage to immune organs during the embryonic stage can lead to immune dysregulation [23,24]. The results of this study showed that, upon *MG* infection, inflammatory damage not only occurred in the respiratory tract and immune organ tissues of chickens but also in those of chicken embryos, especially embryonic spleen; the spleen in two infected birds exhibited inflammatory factor infiltration and lymphocyte reduction. These results suggest that *MG* infection not only crosses the epithelial barrier but may also spread to the immune organs of the chicks and chicken embryos.

Respiratory inflammation induced by *MG* infection was accompanied by activation of immune response involving cytokines production and cell proliferation signal induction [12,15]. It has been reported that immune responses induced by *MG* were age-related in chickens. Younger birds were found to show more serious clinical symptoms than older birds after *MG* infection [25]. The transcriptional changes involved in immune response were stronger in younger birds than in relatively older birds [5]. In the present study, we found that the genes upregulated in chicks were mainly involved in cell proliferation and DNA replication, followed by CCRs interaction and TLR pathway, while genes upregulated in chicken embryos were mostly involved in CCRs interaction and TLR pathways, followed by phagocytosis; the transcriptional changes associated with the immune response were less severe in chicks than in chicken embryos. As an important part of the innate immune system, the transcriptional expression of CCRs and TLRs may be the underlying molecular mechanism of *MG*-induced immune dysregulation.

Since the upregulated genes in *MG*-infected chicks and chicken embryos were significantly involved in the regulation of CCR interaction, we further investigated the genes in this pathway. Eight overlapped CCR genes, including CSF3R, TNFRSF6B, IL20RA, IL8L2, CXCL13L2, IL1B, CCR2, and CCL4, were selected for further study by RT-qPCR. According to previous studies, these CCRs were immune and chemotactic regulators, which closely participated in *MG* infection in chickens [5,26,27]. Consistent with the sequencing data, these CCR genes were highly expressed in the lungs of two birds after *MG* infection, and their expression levels in chicken embryos were higher than those in chicks. Further GO analysis indicates that these CCR genes were involved in neutrophil degranulation and chemotaxis.

TLRs play important roles in inflammation and innate immune response [28,29,30,31]. Given the significant enrichment of TLR signaling in the host response to *MG* infection, we focused on the differences in TLR expression between infected chicks and chick embryos. Although most avian TLRs, which are localized at membrane or endosomal, have been reported to be associated with *MG* infection, especially TLR2, 4, 7, and 15 [5,27], only the role of membrane-localized TLR2, 4, and 6 have been experimentally confirmed. Inhibition of TLR2 suppressed inflammatory cytokines genes induced by *MG* in tracheal epithelial cells [11]. Our group found that in the absence of TLR2, *MG* could induce inflammation via TLR6 in DF-1 cells [32], or TLR4 could be activated by H*MG*B1 to trigger *MG*-induced immune disorders in avian macrophage cells [33]. TLR3, 7, and 21 are localized at endosomal and induced interferon transcription and mixed Th1 and Th2 cytokines responses. However, synergistic TLR21 and TLR3 agonists led to a stronger Th1-biased immune response in chicken monocytes [34]. TLR15 is unique to avian species; upregulation of TLR15 led to activation of the NF-κB pathway and, therefore, pro-inflammatory cytokines production [35,36]. In this study, we found that most TLRs were upregulated in infected chicken embryo lungs but not TLR5 and TLR21. TLR2, 6, and 7 were conservatively upregulated in the lungs of infected chicken embryos and chicks, while TLR3, 4, 5, 15, and 21 showed an opposite expression pattern. Besides, TLR2 and TLR6 were co-upregulated in the immune organs of the two infected birds, except for the chick spleen. Combined with the above findings, TLR2, 6, and 7 may be responsible for *MG*-induced immune dysregulation in chickens. An interesting finding here was that the spleen in infected two birds exhibited relatively lower TLRs expression than the thymus and bursa in response to *MG*. This may be related to lymphocytopenia and inflammatory infiltration caused by *MG* infection in the spleen. Another interesting finding was that TLR3, 7, and 21 were not upregulated simultaneously after *MG* but in pairs. However, this did not include TLR3 and TLR21. For example, TLR3 and TLR7 were co-upregulated in infected chicken embryo lungs, but not TLR21; in infected chick lungs, TLR7 and TLR21 expressions were increased simultaneously, but not TLR3. These may suggest that the immune response to *MG* infection is a mixed Th1 and Th2 cytokines response. Taken together, the higher TLR and CCR expression levels in chicken embryos lead to a stronger innate immune response to *MG* infection in chick embryos than in chicks.

Of interest, *MG* does not contain LPS, but TLR4 expression was increased in infected embryonic lungs in the present work. Owing to the function of activating cytokines and defensins, MMP7 was considered to induce TLR4 upregulation in *MG* infection [12]. A recent study confirmed that the S100A9-coded MRP-126 protein could induce TLR4 expression in the presence of the cofactors CD14 and LY98 [37]. In this study, MMP7 expression was found to be increased 20-fold in chicken lungs and was negatively related to *MG* load in alveolar epithelial cells induced by *MG* but not in chicken embryonic lungs. However, S100A9 was evidently upregulated in the lungs of two infected birds and was also related to *MG* load. Hence, we believe that S100A9 may be responsible for TLR4 activation in *MG* infection, but this needs to be further studied.

Avian AMPs with antibacterial and immunomodulatory activities are important parts of chicken innate and adaptive immune system, which are composed of cathelicidins (CATH) and β-defensins. Unlike most antibiotics, AMPs are less prone to drug resistance due to their unique antibacterial mechanism [38]. It has been reported that the disease-resistant chicken had higher AMPs levels than those in the susceptible chicken [39,40]. In this study, we found nine HDPs, including AVBD1, 4, 5, 6, 7, CATH-1, 2, 3, and DEFB4A, were highly expressed in infected chicken embryo lungs, while only one, AVBD1, was upregulated in infected chick lungs. As in ovo administration has been widely used to improve the immunity of newly hatched chicks [39,41], in ovo administration of AMPs may be a safe and effective way to improve the resistance of poultry to *MG* infection.

Overall, there are certain similarities and differences in the way chicks and chicken embryos coordinate innate responses, ranging from the quantity and quality of the recognition of pathogen receptors, downstream cytokines, and antimicrobial peptides to the mechanism of action. This difference may be due to differences in physiological structure and immune development between chickens and their embryos. Compared to chicks, the lung tissue development of chicken embryos is incomplete, and their tissue development integrity and gas exchange capabilities are inferior to those of chicks. Similarly, the immune system development of chicken embryos is not as complete as that of nursing chicks [7]. Although the immune system of a newborn chick is also immature, it is still better developed than that of an embryo. This means that when the embryo is faced with *MG* invasion, the cytokine and natural antimicrobial peptides-mediated defense mechanisms must be fully activated, resulting in a stronger immune response induced by *MG* in the embryo than in the chick. As many studies have shown that in ovo vaccination can improve the immunity of newborn chicks, the results of this study are beneficial for future exploration and dissection of innate immune signaling pathways, as well as for designing more effective disease control strategies.

## 5. Conclusions

In conclusion, the immune response of chicken embryos to *MG* infection was stronger than that of chicken lungs, which was due to the higher number and expression levels of CCRs, TLRs, and AMPs in chicken embryos. Functional expression of TLRs, CCRs, and AMPs in chicken embryos suggests that immune responses can be modulated and enhanced at this development stage to improve the innate or adaptive immunity of chickens in response to *MG* infection.

## Figures and Tables

**Figure 1 animals-13-01667-f001:**
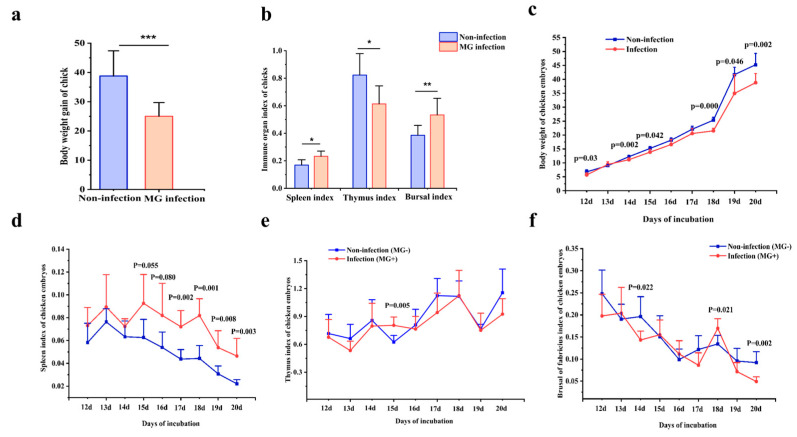
Effect of *MG* infection on body weight and immune organ index in chickens. (**a**) Body weight gains of chicks before and after *MG* infection. (**b**) Organ indexes for spleen, thymus, and bursa in chicks. (**c**) body weight of chicken embryos after 3–11 days of *MG* infection; spleen, thymus, and bursal indexes of chicken embryos after 3–11 days of *MG* infection (**d**–**f**). The data are represented as mean ± SD. * *p* < 0.05; ** *p* < 0.01; *** *p* < 0.001, *n* = 6.

**Figure 2 animals-13-01667-f002:**
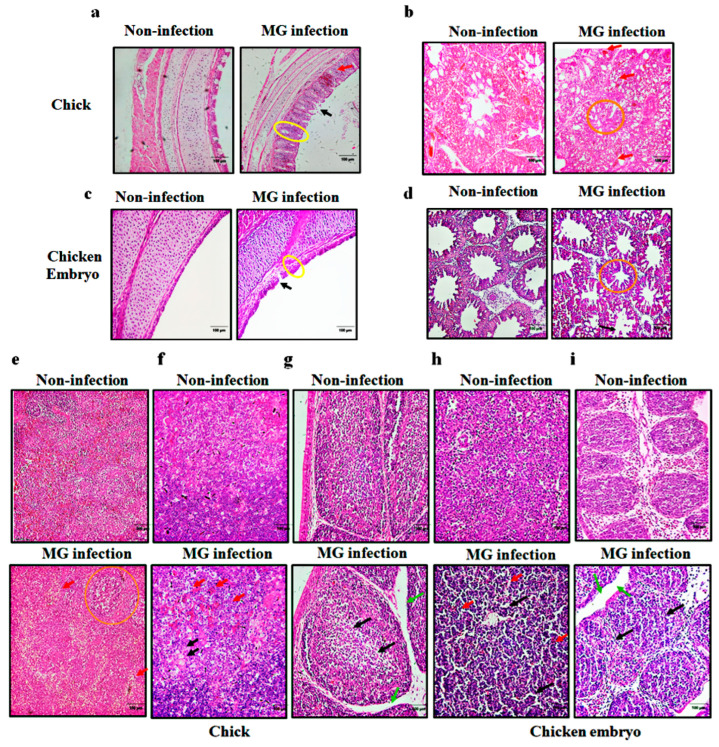
Effect of *MG* infection on histopathology of trachea, lung, and immune organs in chicks and chicken embryos. Histopathological changes of trachea and lungs in chicken embryos (**a**,**b**) and chicks (**c**,**d**). Increased tracheal mucosa thickness (yellow circles); inflammatory cell infiltration (red arrows); an increase in the alveolar space (orange circle); shedding of cilia or epithelial cells (black arrows). (**e**–**i**) histopathological changes of spleen and bursa of Fabricius in chicken embryos (**e**,**f**) and newly hatched chicks (**g**–**i**). Enlarged splenic nodule (orange circle); lymphocyte reduction (black arrows); inflammatory cell infiltration (red arrows); interstitial edema (green arrows). (Magnification: ×20; scale bar: 100 μm; *n* = 3).

**Figure 3 animals-13-01667-f003:**
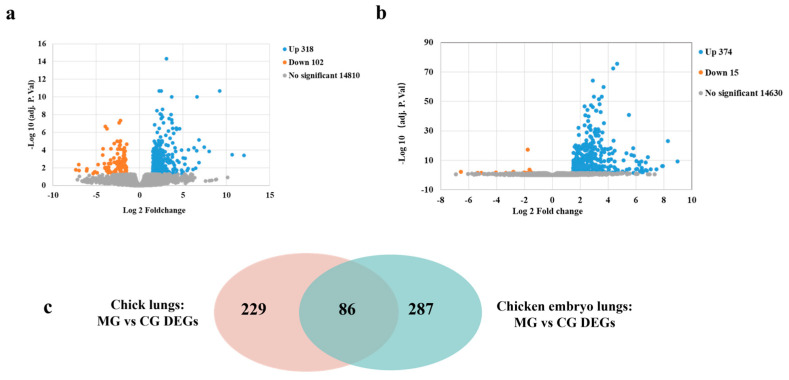
Global DEGs identified between the *MG* infection group and control group in chicks and chicken embryos. Volcano plots of DEGs between the two groups in chicken embryos (**a**) and chicks (**b**). Blue dots indicate significantly up-regulated genes (Log_2_ FC > 1.5 and adjusted-*p* value < 0.05); orange dots indicate significantly down-regulated genes (Log_2_ FC < 1.5 and adjusted-*p* value < 0.05); gray dots indicate insignificant DEGs (adjusted-*p* value > 0.05). (**c**) Venn diagram of identified DEGs that are identical or specific between chicks and chick embryos with *MG* infection.

**Figure 4 animals-13-01667-f004:**
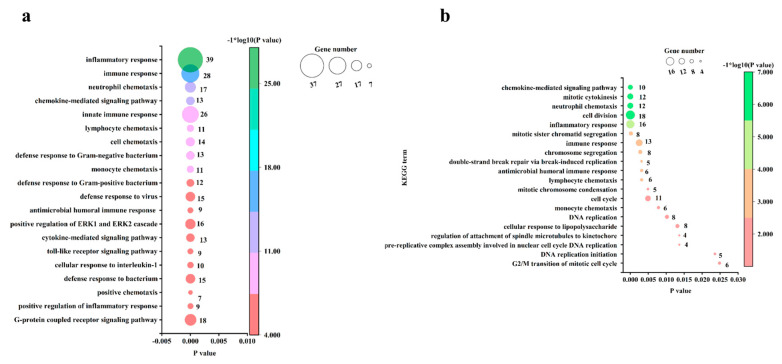
The most significant enriched BPs based on FDR. (**a**) BPs enriched with up-regulated genes in *MG*-infected chicken embryo lungs compared with the control group. (**b**) BPs enriched with upregulated genes in *MG*-infected chick lungs compared with the control group. BP terms enriched at an FDR < 0.05 were listed.

**Figure 5 animals-13-01667-f005:**
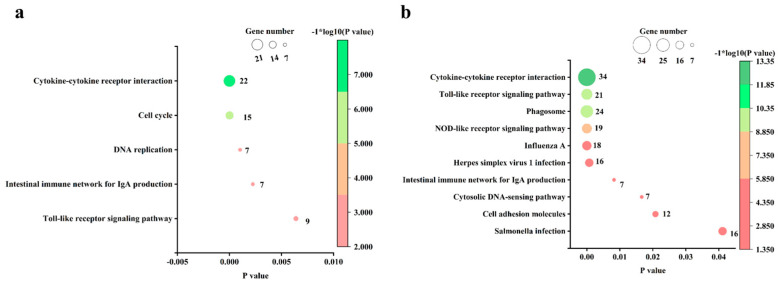
The significantly enriched KEGG pathways based on FDR. (**a**) KEGG pathways enriched in upregulated genes in *MG*-infected chicken embryo lungs compared to the control group. (**b**) KEGG pathways enriched in upregulated genes in *MG*-infected chick lungs compared to the control group. KEGG terms enriched at an FDR < 0.05 were listed.

**Figure 6 animals-13-01667-f006:**
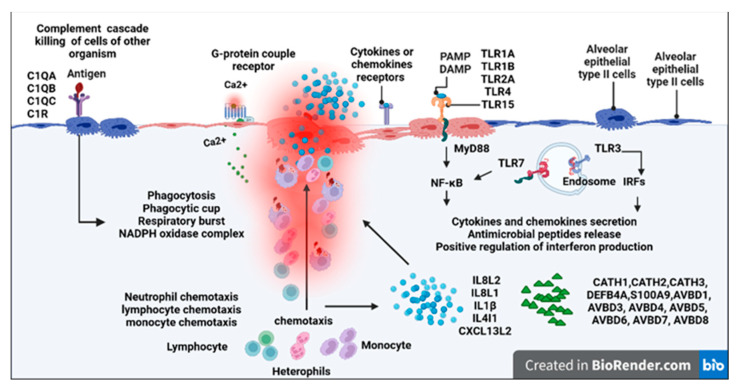
Chicken embryo lungs responses to *MG* infection. The text indicates genes, gene ontologies, protein classes, and pathways enriched with upregulated genes. Created with BioRender.com (accessed on 1 February 2023).

**Figure 7 animals-13-01667-f007:**
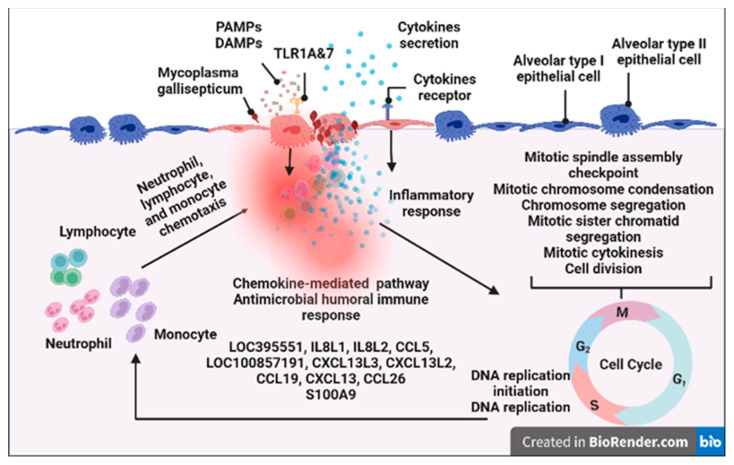
Newly hatched chick lungs responses to *MG* infection. The text indicates genes, gene ontologies, protein classes, and pathways enriched with upregulated genes. Created with BioRender.com (accessed on 1 February 2023).

**Figure 8 animals-13-01667-f008:**
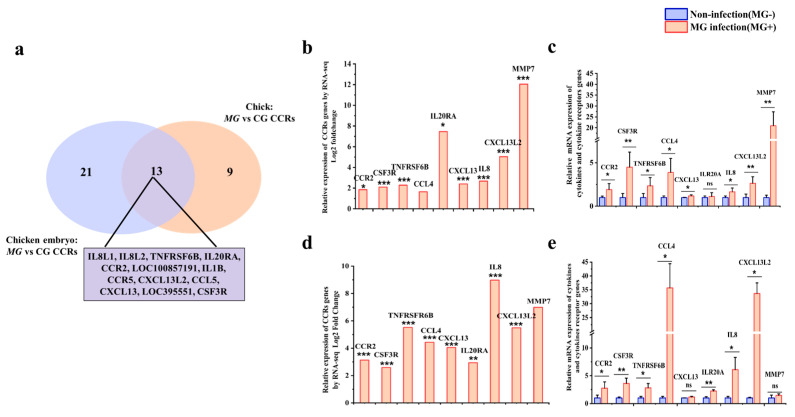
Effects of *MG* infection on CCRs genes expressions in chicken embryos and chicks. (**a**) Venn diagram of DEGs in the CCRI pathway in infected chicken embryo and chick lung tissue. (**b**) The mRNA expression level of CCRs and MMP7 in lungs of chicken embryos after *MG* infection by RNA-seq. (**c**) The mRNA expression level of CCRs and MMP7 in lungs of chicken embryos after *MG* infection by RT-qPCR. (**d**) The mRNA expression level of CCRs and MMP7 in lungs of chicks after *MG* infection by RNA-seq. (**e**) The mRNA expression level of CCRs and MMP7 in lungs of chicks after *MG* infection by RT-qPCR. GAPDH was used as a control. The data were represented as the mean ± SD. *n* ≥ 4. * *p <* 0.05; ** *p* < 0.01, *** *p* < 0.001.

**Figure 9 animals-13-01667-f009:**
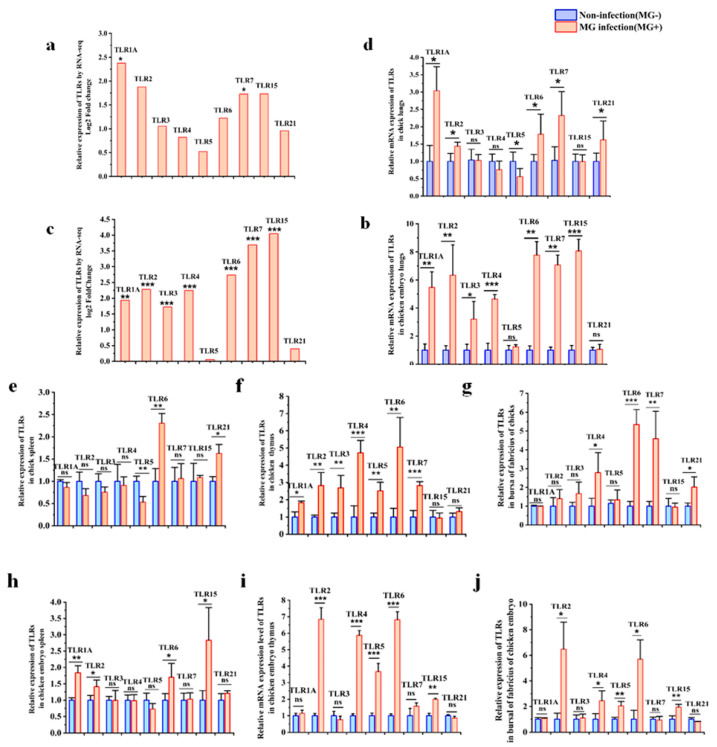
Effects of *MG* infection on TLRs gene expressions in chicken embryos and chicks. (**a**) The mRNA expression level of TLRs in lungs of chicken embryos after *MG* infection by RNA-seq. (**b**) The mRNA expression level of TLRs in lungs of chicken embryos after *MG* infection by RT-qPCR. (**c**) The mRNA expression level of TLRs in lungs of chicks after *MG* infection by RNA-seq. (**d**) The mRNA expression level of TLRs in lungs of chicks after *MG* infection by RT-qPCR. The mRNA expression level of TLRs in the spleen (**e**), thymus(**f**), and bursa of Fabricius (**g**) of chicken embryos after *MG* infection by RT-qPCR. The mRNA expression level of TLRs in the spleen (**h**), thymus (**i**), and bursa of Fabricius (**j**) of chicks after *MG* infection by RT-qPCR. GAPDH was used as a control. The data were represented as the mean ± SD. *n* ≥ 4. * *p* < 0.05; ** *p* < 0.01, *** *p* < 0.001.

**Figure 10 animals-13-01667-f010:**
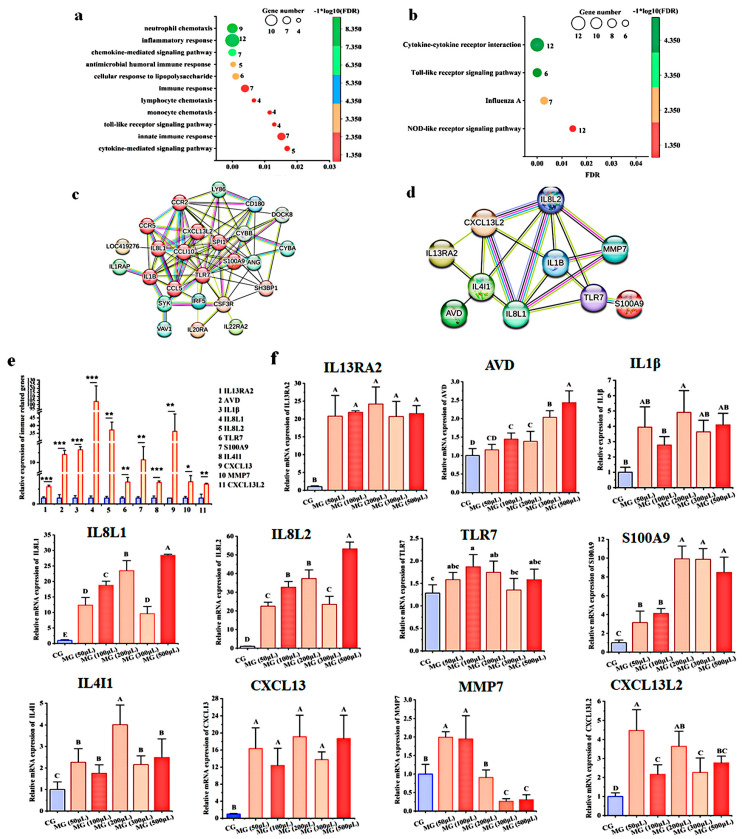
Analysis of protein interaction network (**a**) BPs enriched with shared upregulated genes by *MG*-infected chicken embryo and chick lungs. (**b**) KEGG pathways enriched with shared upregulated genes by *MG*-infected chicken embryo and chick lungs. GO and KEGG pathway terms enriched at an FDR of <0.05 were listed. (**c**) STRING analysis of the protein interaction network of shared upregulated genes by *MG*-infected chicken embryo and chick lungs. (**d**) STRING analysis of the protein interaction network of core immune genes. (**e**) The mRNA expression levels of core immune genes in *MG*-infected AEC-II cells. (**f**) The effect of *MG* concentration on the mRNA expressions of core immune genes. GAPDH was used as a control. The data were represented as the mean ± SD. *n* ≥ 4. * *p* < 0.05; ** *p* < 0.01; *** *p* < 0.001. Bars with different capital letters show a significant difference (*p <* 0.01), and bars with different lowercase letters show a significant difference (*p* < 0.05).

**Table 2 animals-13-01667-t002:** DNA sequence of primers.

Name	Primers	Accession No.
TLR1B-F	GCCGTTGCCTTCTAAAAGAGG	NM_001081709.4
TLR1B-R	CCAGGACACTCTGACAGGGA	NM_001081709.4
TLR2A-F	AAATCAGCGGGATGCACCTA	NM_001396827.1
TLR2A-R	TTGGCATCGGATCACAGGTC	NM_001396827.1
TLR7-F	TCTTTCAGAGGTGGCTGCAC	XM_046908011.1
TLR7-R	GATCCCTCTGGGGACTTCCT	XM_046908011.1
TLR4 -F	ATCTTTCAAGGTGCCACATC	NM_001030693.2
TLR4 -R	GGATATGCTTGTTTCCACCA	NM_001030693.2
TLR3-F	CTGCGGAATCTGACTGTCCT	NM_001011691.4
TLR3-R	TCCTCCTGGGTTTGCACATTT	NM_001011691.4
TLR5-F	CACCTGTGCCAAAGGACACT	NM_001398059.1
TLR5-R	AAACGTTGCAAACCCGCAAA	NM_001398059.1
TLR15-F	GGCTGTGGTATGTGAGAATG	NM_001398238.1
TLR15-R	ATCGTGCTCGCTGTATGA	NM_001398238.1
TLR21-F	CAGGGTATGCAGCTGTGTCA	NM_001030558.3
TLR21-R	CAGGGTATGCAGCTGTGTCA	NM_001030558.3
CSF3-F	ACCACGACTTCCAGCTCTTC	NM_205279.2
CSF3-R	CTGGAAGGTGTCACACACGG	NM_205279.2
CSF3R-F	TCCTCACACTCCCTCTTTGC	NM_001030898.2
CSF3R-R	ATGGGATGGTGCCTTCTGC	NM_001030898.2
TNFRSF6B-F	AGTGCCTCTACTGCAACGTC	XM_417434.6
TNFRSF6B-R	CAGAGGGGGAACCCAACTTC	XM_417434.6
IL-1*β*-F	TGCCTGCAGAAGAAGCCTCG	NM_204524.1
IL-1*β*-R	CTCCGCAGCAGTTTGGTCAT	NM_204524.1
CCL4-F	AGCGTAGGAACTCCACTCTC	XM_015295666.2
CCL4-R	CCTTCTTTGTGATGAGATGATGGC	XM_015295666.2
CXCL13L2-F	CAACTGCCTTCATTCCCTTGC	NM_001348656.1
CXCL13L2-R	GCAGCTTTCTTCTTCAGCTTCG	NM_001348656.1
CXCL13-F	TGTCAAAGTGACTGCCCAGA	NM_001348657.1
CXCL13-R	CCTCTTCAGGGTGAGGATGATCT	NM_001348657.1
IL20RA-F	CGGCGCACACAGGAATGTT	XM_015284185.2
IL20RA-R	GCTCTTCATGGTCAGCAGTCT	XM_015284185.2
IL8L1-F	ATGTGAACCTCACCCCTAGC	NM_205018.1
IL8L1-R	TGAATGGCGTTGTCTCCCAC	NM_205018.1
IL13RA2-F	GGAGGTCCAGAGTGAATGGC	NM_001048078.1
IL13RA2-R	AGCATGCAGCCCATGTTTAC	NM_001048078.1
IL1R2-F	ATTTCCGCTGTGCCTTGAGT	XM_015277810.2
IL1R2-R	CAGTAGGAGTTGTTCCTGTGCT	XM_015277810.2
MMP7-F	CAGAGCCTTCCGTTTCCAGT	NM_001006278.1
MMP7-R	GGGATTCCACATCTGGGCTG	NM_001006278.1
OLFM4-F	AGAAGACTACCAGCCAGCAC	NM_001040463.2
OLFM4-R	AAAGGTGGTATCCGGCAAGT	NM_001040463.2
AVD-F	CCCACCTTTGGCTTCACTG	NM_205320.2
AVD-R	ACCTCCTTCCCGTTCCTGT	NM_205320.2
IL8L2-F	TTCAGCTGCTCTGTCGCAA	NM_205498.2
IL8L2-R	GCACACCTCTCTTCCATCCTT	NM_205498.2
S100A9-F	CCACTTCTGTGAGGACCACC	NM_001305151.2
S100A9-R	GGCAGCAAAACCATCAAACCT	NM_001305151.2

**Table 3 animals-13-01667-t003:** Top 30 common upregulated genes in the lung tissues of *MG*-infected chicken embryos and newly hatched chicks.

Gene Name	Chicken Embryos CG vs. *MG*		Newly Hatched Chicks CG vs. *MG*	
	Log_2_ FC	*p* Value-Adj	Log_2_ FC	*p* Value-Adj
LOC770026	7.871279	5.93 × 10^−7^	5.628906	9.02 × 10^−5^
IL4I1	8.292192	9.01 × 10^−24^	4.34217	0.001511
IL8L2	8.973782	4.22 × 10^−10^	2.677474	0.000107
AVD	8.112968	1.10 × 10^−229^	3.530443	0.001406
CXCL13	4.050071	2.39 × 10^−4^	7.463658	4.76 × 10^−5^
IL8L1	6.075514	5.63 × 10^−5^	4.837175	0.01823
CXCL13L2	5.491891	1.62 × 10^−41^	5.034946	0.000766
CLEC5A	6.426947	1.13 × 10^−5^	3.781138	0.000599
DCSTAMP	5.106985	8.03 × 10^−11^	4.602756	3.74 × 10^−7^
CD72L1	5.84905	7.68 × 10^−14^	3.81117	1.56 × 10^−5^
MLKL	6.022089	2.49 × 10^−10^	3.513237	0.002472
CD72	4.639555	5.09 × 10^−3^	4.272585	4.67 × 10^−7^
MRGPRH	4.359756	4.33 × 10^−73^	4.054451	0.010629
S100A9	4.645942	2.61 × 10^−76^	3.256855	0.001827
LOC419276	4.625564	3.41 × 10^−3^	2.765353	0.025887
MX1	4.17664	5.01 × 10^−19^	2.5511	0.00012
LOC101748032	3.504725	4.38 × 10^−14^	3.160072	0.001962
LPAR5	3.489633	0.023681	2.833963	0.014503
CCL5	3.601561	9.63 × 10^−7^	2.597259	0.018252
CYBB	3.470943	1.64 × 10^−43^	2.539925	0.005234

**Table 4 animals-13-01667-t004:** Similarities and differences of immune damage induced by *MG* infection on chicks and chicken embryos.

Feature	Pathological Changes/Enriched GO and KEGG Term	Affected Genes
Similarities	(1) decreased body weight	No
	(2) increased spleen and thymus organ indices	No
	(3) increased tracheal mucosal thickness and narrowed alveolus space with inflammatory infiltration in the trachea and lungs	No
	(4) Enrichment of GO and KEGG pathways involved in TLRs	Upregulation of genes for TLR6, TLR7, and TLR1A in lung tissues; up-regulation of genes for TLR6 in spleen, thymus, and bursa of Fabricius;
	(5) Enrichment of GO and KEGG pathways involved in cytokines and chemokines	Upregulation of genes for IL8L1, IL8L2, TNFRSF6B, IL20RA, CCR2, LOC100857191, IL1B, CCR5, CXCL13L2, CCL5, CXCL13, LOC395551, and CSF3R;
	(6) Enrichment of GO terms involved in antimicrobial peptide-mediated immune response	RNASE6, S100A9, RSFR, and LBP.
	(7) Enrichment of GO terms involved in neutrophil degranulation: dysregulation	IL1B, HBE1, RNASE6, S100A9, BST1, ADGRG5, RSFR, LBP, and CXCL13L2;
	(8) Enrichment of GO and KEGG pathways involved in neutrophil chemotaxis	IL8L1, CSF3R, IL8L2, SYK, CCL5, LOC100857191, CXCL13L2, CXCL13, CCL26;
Differences	(1) The number of DEGs related to TLRs in chicks was lower than that in chicken embryos	Upregulations of genes for MAP3K8, SPP1, TLR4, TLR2A, TLR15, CD88, FOS, CCL4, TLR6, TLR3, LY96, and IRF7 were observed in chicken embryos, but not chicks
	(2) The number of DEGs related to cytokines, chemokines, and their receptors in chicks was lower than that in chicken embryos	Upregulations of genes for CSF3, IL18RAP, CSF2RB, IL16, IL18R1, IL10RA, CCL20, CSF1R, IL18, IL13RA2, TNFRSF4, IL1R2, XCR1, IL2RA, CCL4, CX3CR1, CXCR1, CRCBL, IL2RG, CSF2RA, and LOC101747500 in were observed in chicken embryos, but not in chicks
	(3) The number of DEGs related to antimicrobial peptides in chicks was lower than that in chicken embryos	Upregulations of AVBD1, 3, 4, 5, 6, 7, and 8 and DEFB4A, CATH1, CATH2, and CATH3 were observed in chicken embryos only
	(4) Enrichment of GO and KEGG pathways involved in phagocytosis in chickens but not in chicks	Upregulations of C1R, NCF2, BLB1, NCF4, ITGB2, TUBA4AL, TLR1A, TAP1, NCF1C, CYBB, TUB4A, TLR2A, BF1, MMR1L3, LOC771876, MMR1L4, MARCO, DMA, DMB2, LOC420160, ATP6V1G3, ATP6V0D2, TLR4, LOC428421 were observed in chicken embryos only
	(5) Enrichment of GO and KEGG pathways involved in DNA replication and cell cycle in chicks, but not in chicken embryos	Upregulations of FEN1, RFC3, MCM3, MCM4, MCM5, DNA2, MCM2, BUB1B, CDK1, BUB1 were observed in chicks only

Notes: The similarities and differences between chick and chicken embryos listed in the table were mainly drawn based on GO and KEGG enrichment analysis results.

## Data Availability

The data supporting the conclusions of this study are available from the corresponding author upon reasonable request.

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
