# Peer review of "Comparative Transcriptome Analysis Reveals the Innate Immune Response to Mycoplasma gallisepticum Infection in Chicken Embryos and Newly Hatched Chicks"

_animals, 2023, doi:10.3390/ani13101667_

Round 1

Reviewer 1 Report

This study provides valuable insights into the innate immune response of chicken embryos and newly hatched chicks to MG infection and insightful findings on the impact of MG infection on chicken health. It is a very well carried out work, from the introduction to the discussion, with a cutting-edge methodology. This article is well-written and clearly presents the research objectives, methods, and key findings, making it a valuable resource for researchers and practitioners interested in poultry health and disease control. However, the article requires improvements in the following areas to meet publication standards.

Minor concerns:

1) Upon comprehensive examination of the article, I believe that a more fitting title would be "Comparative transcriptome analysis of innate immune response to Mycoplasma gallisepticum infection in chicken embryos and newly hatched chicks," as the study did not extensively investigate the TLR7 signaling pathway.

2) It is suggested that the author explain a rationale for the use of alveolar type II cells research over other types of chicken cells(such as DF-1 cells) in the introduction.

3) There are some grammatical and spell errors in this article, please carefully review and rectify any grammatical or coherence errors present throughout the article to bring it up to a publishable standard.

Such as:

Line93-95: At 5 days post-infection, the chicks in the infected group all showed clinical symptoms of rales, mouth breathing, coughing and sneezing and were humanely euthanized for body weight detection and tissue collection.

Line 361-364: Protein interaction network analysis through STRING showed that 10 genes including but not limited to TLR7, S100A9, IL8L1, and IL1B were located in the center of the in-teraction network, indicating that these genes may play dominant roles in response to MG infection in both chicken embryos and chicks (Fig 8c).

Line407: The spleen contains a large number of lymphocytes and is the center of cellular and humoral immunity [17].

Line 443- 444“Although most avian TLRs, which localized at membrane or endosomal, have been reported to be associated with MG infection, especially TLR2, 4,7, 15 [5, 6, 22], only the role of membrane-localized TLR2, 4, 6 have been experimentally confirmed”.

Line 446-447: Our group found that, in the absence of TLR2, MG could induced inflammation via TLR6 in DF-1 cells [23], or TLR4 could be activated by HMGB1 to trigger MG-induced immune disorders in avian macrophages cells [24].

Author Response

Thanks for the time and effort you spent in reviewing our manuscript and providing constructive comments. Based on your comments, we have made careful revisions on the manuscript, which has significantly improved the quality of the manuscript. Should you have any question or additional comments, please feel free to let us know. Our point-by-point responses are provided as below

Response to reviewer 1

  • Upon comprehensive examination of the article, I believe that a more fitting title would be "Comparative transcriptome analysis of innate immune response to Mycoplasma gallisepticum infection in chicken embryos and newly hatched chicks," as the study did not extensively investigate the TLR7 signaling pathway.

Response: Thank you for taking the time to thoroughly review our article and for your helpful suggestion regarding the title. We appreciate your input and have revised the title to better reflect the focus of our study.

  • It is suggested that the author explain a rationale for the use of alveolar type II cells research over other types of chicken cells(such as DF-1 cells) in the introduction.

Response: Thank you for your valuable feedback. We have added a brief explanation in the introduction section to clarify the rationale for using alveolar type II cells in our research instead of other chicken cells. We appreciate your suggestion and will make the necessary changes to improve the clarity of our study.

  • There are some grammatical and spell errors in this article, please carefully review and rectify any grammatical or coherence errors present throughout the article to bring it up to a publishable standard.

Line93-95: At 5 days post-infection, the chicks in the infected group all showed clinical symptoms of rales, mouth breathing, coughing and sneezing and were humanely euthanized for body weight detection and tissue collection.

Line 361-364: Protein interaction network analysis through STRING showed that 10 genes including but not limited to TLR7, S100A9, IL8L1, and IL1B were located in the center of the in-teraction network, indicating that these genes may play dominant roles in response to MG infection in both chicken embryos and chicks (Fig 8c).

Line407: The spleen contains a large number of lymphocytes and is the center of cellular and humoral immunity [17].

Line 443- 444“Although most avian TLRs, which localized at membrane or endosomal, have been reported to be associated with MG infection, especially TLR2, 4,7, 15 [5, 6, 22], only the role of membrane-localized TLR2, 4, 6 have been experimentally confirmed”.

Line 446-447: Our group found that, in the absence of TLR2, MG could induced inflammation via TLR6 in DF-1 cells [23], or TLR4 could be activated by HMGB1 to trigger MG-induced immune disorders in avian macrophages cells [24].

Response: Thank you for your helpful feedback. We will carefully review our article to identify and rectify any grammatical or coherence errors to bring it up to a publishable standard. We appreciate your input and will ensure that our article is error-free and meets the required standard before submission.

Reviewer 2 Report

This paper compares immune gene expression of SPF chicken embryos and chicks infected with Mycoplasma gallisepticum. The paper is well-written and of interest to infectious disease researches and immunologists. The results are interesting and are presented clearly; there are only minor syntax changes required. However, there are a few issues that need to be addressed. Please refer to my queries below for revision of the manuscript. 

Line 46: Should read “The chicken embryo…”

Line 47: Should read “…vaccine efficacy.”

Line 78: The 6 in 1 x 106 should be a superscript.

Line 85: In addition to the provided references, please consider making a brief statement that specifies the source of SPF eggs, total number of eggs infected and/or eggs per treatment group, embryonic day of infection, method of infection, and any modifications from the referenced paper.

Line 90: In addition to the provided references, please consider making a brief statement that specifies the source of SPF chicks, total number of chicks infected and/or chicks per treatment group, day of infection, method of infection, and any modifications from the referenced paper.

Line 93: F in “Fresh spleen…” should be lowercase.

Line 104: Should read “All DNA primer sequences…”

Line 124:  Should say either reported or considered, not both words.

Line 125: It appears that 2 birds per group were sampled for histopathology. Please specify the number of samples allocated for these procedures, assuming the other randomly collected samples were used for total RNA isolation and sequencing.  

Line 185: The orange circles depict decreased alveolar space, not increased.

Table 3: This similarities/differences table is an important summary of findings. However, it needs to be reformatted for reader clarification. Firstly, the title refers only to lung changes however decreased body weight and organ indices are not related to the lung. This could be mitigated by making the title less specific such as “Similarities and differences between immunopathological changes induced by MG infection…” Further, the differences section 1 – 4 has two columns, which is not consistent with the similarities section. This section also has no reference for the differences. For example, in differences section 1, are the embryo or chick lungs more dependent on TLR activation? This can be inferred by reading the results but tables should be standalone and able to be interpreted without the context of the text. My suggestion would be to add a solid line between similarities section 7 and differences section 1 to show the divide and divide the content into two columns: one for GO feature (e.g. neutrophil degranulation) and another for genes affected.

Line 397: While the spleen is important for avian immune function, I disagree that the spleen is the center of cellular and humoral immunity. The spleen is a peripheral immune organ in birds while the bone marrow, bursa, and thymus are considered central immune organs. However, it does have an important role in the embryo in that B cells undergo rearrangement in the spleen prior to maturation in the bursa. This may be relevant to the point you are trying to make. Please refer to and cite avian-specific literature (https://doi.org/10.1186/s40104-021-00559-1 and https://doi.org/10.1016/B978-0-12-407160-5.00017-8) and rephrase this sentence and the section that follows.

Line 401: Should read “two infected birds”

Line 407: Should read “Younger birds were found to show…,” or “Younger birds showed…”

Line 438: Should read “TLR15 is unique to avian species, and…”

Overall comment: Due to the comparative aspect of this manuscript (hatched chicken vs embryo), a brief summary of immune development in the chicken is warranted. This will add context to the changes detected in the sampled organs. Further, I would add a statement about how the embryonic lung and spleen differs from the that of a hatched chicken, both in physiology and immune function, and how that may have affected the results. 

Author Response

Manuscript ID animals-2287446 entitled “TLR7 Mediates the Innate Immune Responses in Mycoplasma gallisepticum Infection”

Thanks for the valuable time and effort you spent in reviewing our manuscript and providing constructive comments. We are pleased that you found our study to be well carried out and valuable to the poultry health and disease control community. We have carefully considered your suggestions and make the necessary improvements to ensure that our article meets publication standards. Thank you again for your review and input. Should you have any question or additional comments, please feel free to let us know. Our point-by-point responses are provided as below

Response to reviewer 2

  1. Line 46: Should read “The chicken embryo…”

Response: Thank you your careful review. We have made the necessary correction to line 46 to ensure accuracy and clarity.

  1. Line 47: Should read “…vaccine efficacy.”

Response: We appreciate your careful review and have made the necessary correction to line 47 to ensure accuracy and clarity.

Line 78: The 6 in 1 x 106 should be a superscript.

Response: We appreciate your careful review and have made the necessary correction to line 78 to ensure accuracy and clarity.

  1. Line 85: In addition to the provided references, please consider making a brief statement that specifies the source of SPF eggs, total number of eggs infected and/or eggs per treatment group, embryonic day of infection, method of infection, and any modifications from the referenced paper.

Response: Thank you for your helpful suggestion. We have provided additional information as suggested regarding the source of SPF eggs, total number of infected eggs and/or eggs per treatment group, embryonic day of infection, method of infection, and any modifications made from the referenced paper. We appreciate your input and will ensure that our article provides sufficient information for readers to understand our methodology.

  1. Line 90: In addition to the provided references, please consider making a brief statement that specifies the source of SPF chicks, total number of chicks infected and/or chicks per treatment group, day of infection, method of infection, and any modifications from the referenced paper.

Response: Thank you for your helpful suggestion. We have provided additional information regarding the source of SPF chicks, total number of infected chicks and/or chicks per treatment group, day of infection, method of infection, and any modifications made from the referenced paper. We appreciate your input and will ensure that our article provides sufficient information for readers to understand our methodology.

Revised version:

  1. Line 93: F in “Fresh spleen…” should be lowercase.

Response: Thanks very much for your careful review. We have corrected this mistake.

  1. Line 104: Should read “All DNA primer sequences…”

Response: Thanks very much for your careful review. We have corrected this mistake.

  1. Line 124:  Should say either reported or considered, not both words.

Response: Thanks very much for your careful review. We have deleted the word “reported”.

  1. Line 125: It appears that 2 birds per group were sampled for histopathology. Please specify the number of samples allocated for these procedures, assuming the other randomly collected samples were used for total RNA isolation and sequencing.  

Response: Thanks very much for your careful review. There are 3 bird per group were sampled for histopathology and sequencing and we have added this information in Material and methods section.

  1. Line 185: The orange circles depict decreased alveolar space, not increased.

Response: Thanks very much for your careful review. We corrected this mistake.

  1. Table 3: This similarities/differences table is an important summary of findings. However, it needs to be reformatted for reader clarification. Firstly, the title refers only to lung changes however decreased body weight and organ indices are not related to the lung. This could be mitigated by making the title less specific such as “Similarities and differences between immunopathological changes induced by MG infection…” Further, the differences section 1 – 4 has two columns, which is not consistent with the similarities section. This section also has no reference for the differences. For example, in differences section 1, are the embryo or chick lungs more dependent on TLR activation? This can be inferred by reading the results but tables should be standalone and able to be interpreted without the context of the text. My suggestion would be to add a solid line between similarities section 7 and differences section 1 to show the divide and divide the content into two columns: one for GO feature (e.g. neutrophil degranulation) and another for genes affected.

Response: Thank you for your valuable feedback regarding Table 3. We agree that the title should be less specific to better reflect the content of the table and we have changed the title of the table to “Similarities and differences of immune damage induced by MG infection on chicks and chicken embryos”

We appreciate your suggestions and have added a solid line between the similarities and differences sections and divided the content in similarities into two columns to improve clarity. Additionally, we have revised the differences section to ensure that readers can interpret the table without the context of the text.

  1. Line 397: While the spleen is important for avian immune function, I disagree that the spleen is the center of cellular and humoral immunity. The spleen is a peripheral immune organ in birds while the bone marrow, bursa, and thymus are considered central immune organs. However, it does have an important role in the embryo in that B cells undergo rearrangement in the spleen prior to maturation in the bursa. This may be relevant to the point you are trying to make. Please refer to and cite avian-specific literature (https://doi.org/10.1186/s40104-021-00559-1IF: 6.175 Q1 and https://doi.org/10.1016/B978-0-12-407160-5.00017-8)and rephrase this sentence and the section that follows.

Response: Thank you for your valuable feedback regarding our article. We couldn’t agree more that the spleen is a peripheral immune organ in birds while the bone marrow, bursa, and thymus are considered central immune organs, we apologize for our misrepresentation. We appreciate your suggestion to rephrase the sentence regarding the spleen's role in avian immune function and to refer to and cite avian-specific literature. We have made the necessary changes and ensure that the updated section is clear and accurate. Thank you for bringing this to our attention and helping us improve the quality of our article.

  1. Line 401: Should read “two infected birds”

Response:  Thanks for your careful review. We have corrected this mistake.

  1. Line 407: Should read “Younger birds were found to show…,” or “Younger birds showed…”

Response: Thanks for your careful review. We have corrected this mistake.

  1. Line 438: Should read “TLR15 is unique to avian species, and…”

 Response: Thanks for your careful review. We have corrected this mistake.

  1. Overall comment: Due to the comparative aspect of this manuscript (hatched chicken vs embryo), a brief summary of immune development in the chicken is warranted. This will add context to the changes detected in the sampled organs. Further, I would add a statement about how the embryonic lung and spleen differs from the that of a hatched chicken, both in physiology and immune function, and how that may have affected the results. 

Response: Thank you for your constructive feedback. We appreciate your suggestion and agree that adding a brief summary of immune development in the chicken would help provide context for the results. We have added a brief summary in the discussion section

According to your suggestion, we have added a statement on the differences between the embryonic and hatched chicken lung and spleen physiology and immune function, and how these differences may have influenced the observed results in Discussion section.
